# Non-Ischemic Myocardial Fibrosis in End-Stage Kidney Disease Patients: A New Perspective

**Kenji Nakata and Nobuhiko Joki \***

Department of Nephrology, Toho University Ohashi Medical Center, Tokyo 153-8515, Japan
* Correspondence: jokinobuhiko@gmail.com; Tel.: +81-3-3468-1251; Fax: +81-3-5433-3067

**Abstract:** Cardiovascular medicine, especially for ischemic heart disease, has evolved and advanced over the past two decades, leading to substantially improved outcomes for patients, even those with chronic kidney disease. However, the prognosis for patients with end-stage kidney disease (ESKD) has not improved so greatly. Recent studies have reported that myocardial fibrosis in chronic kidney disease patients is characterized by patchy and interstitial patterns. Areas of fibrosis have been located in the perivascular space, and severe fibrotic lesions appear to spread into myocardial fiber bundles in the form of pericellular fibrosis. These findings are fully consistent with known characteristics of reactive fibrosis. In hemodialysis patients, a greater extent of myocardial fibrosis is closely associated with a poorer prognosis. In this review, we focus on non-ischemic cardiomyopathy, especially reactive myocardial fibrosis, in ESKD patients.

**Keywords:** reactive fibrosis; type 2 myocardial infarction; biomarker

## 1. Introduction

About 20 years have passed since chronic kidney disease (CKD) was first recognized as a contributor to atherosclerotic cardiovascular disease [1,2]. This condition is considered to correspond to group 4 in the CRS classification [3]. The pathogenesis of CRS4 has been predominantly explained by narrowing of the artery and tissue ischemia due to atherosclerosis, as typically seen in coronary artery disease. Cardiovascular medicine, especially for atherosclerosis, has evolved and improved over the past two decades, even for patients with CKD, however its effect is limited. The short-term prognosis after myocardial infarction in CKD patients has improved, however long-term prognosis has not yet improved [4].

The pathophysiology may change as CKD progresses. In CKD 1–3, about 60% of myocardial infarctions are ST-segment elevation myocardial infarction, whereas in CKD 5, about 60% are non-ST-segment elevation myocardial infarction [5]. This implies that in advanced CKD, myocardial injury due to non-atherosclerotic conditions may occur. It is well known that lipid-lowering therapies appear much less effective in patients with advanced stages of CKD, including end-stage kidney disease (ESKD) [6–8], compared with early-stage CKD patients. On the other hand, vascular dysfunction such as arterial distensibility and stiffness is present in the early stage of CKD, and plays an important mediator in the chronic impairment of cardiac function [9]. It has been reported that left ventricular remodeling occurs with the progression of CKD and is often complicated by cardiac hypertrophy [10,11]. It has also been reported that fibrosis of the myocardium appears early in the course of CKD [12,13]. To establish a better prognosis for CKD patients, we must first understand the mechanism and pathophysiology of non-ischemic cardiomyopathy in CKD. In this review, we focus on non-ischemic cardiomyopathy, especially cardiac fibrosis, in end-stage kidney disease (ESKD) patients.

## 2. Myocardial Fibrosis in CKD

Research into non-atherosclerotic pathology has focused on myocardial fibrosis. Aoki et al. performed a coronary angiography on 40 maintenance hemodialysis patients with

left ventricular systolic dysfunction (ejection fraction <50%) and dilated cardiomyopathy. They checked for the absence of significant coronary artery narrowing lesions, and then carried out myocardial biopsies for the purpose of histological study [14]. Fifty patients with dilated cardiomyopathy who were not on dialysis were enrolled as controls. The area of fibrosis in cardiomyocyte cells was found to be 22% in both groups. However, severe fibrosis, defined as an area of 45% or more, was often observed in the dialysis patients compared with the controls. Areas of fibrosis of 30% or more were associated with significantly poorer prognoses in the dialysis group, but not in the control group. Histologically, cardiac fibrosis is broadly categorized into two types: replacement fibrosis and reactive fibrosis, also known as diffuse myocardial fibrosis because of its manifestation as a diffuse and excessive deposition of extracellular matrix components in interstitial and perivascular areas (Figure 1).

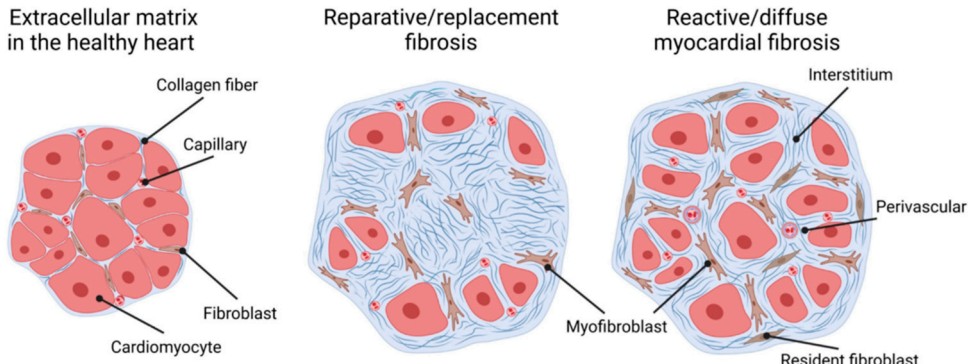

**Figure 1.** Types of cardiac fibrosis [15]. The extracellular matrix in the healthy heart (**left**) is a three-dimensional network of collagen fibers that embeds cardiac cells such as cardiomyocytes, capillaries, and fibroblasts. "Replacement/reparative fibrosis" (**middle**) is visible as a collagen-based scar that is formed during a healing process and replaces dying cardiomyocytes after ischemic insults. "Reactive/diffuse myocardial fibrosis" (**right**) manifests as diffuse deposition of cross-linked collagens in interstitial and perivascular areas.

Replacement fibrosis is the result of a healing process after acute myocardial infarction, or the result of cardiomyocyte death brought about by other causes. It is visible as a fibrotic scar triggered by ischemic cell death. Reactive fibrosis is caused by the diffuse deposition and cross-linking of collagens in interstitial and perivascular areas that occur in chronic cardiac conditions [15–17]. Researchers have considered non-ischemic myocardial fibrosis as expressing the pathological pattern of reactive fibrosis, and it has been observed in hypertensive cardiomyopathy [18], diabetic cardiomyopathy [19], aortic valve stenosis [20], and hypertrophic cardiomyopathy [21].

Diabetic kidney disease is the greatest cause of dialysis-dependent ESKD [22,23]. In addition, CKD patients often exhibit left ventricular hypertrophy [24], which may be due to compensatory mechanisms of pressure overload [25]. The process of CKD progression is now thought to be part of the progression process of reactive cardiac fibrosis. One review article concerning cardiovascular disease clearly states that the myocardial fibrosis in patients with CKD is characterized by patchy fibrosis [26], which is consistent with known characteristics of reactive fibrosis. Izumaru et al. conducted a histological study of 334 autopsy cases [27] to evaluate the myocardial cell width and areas of myocardial fibrosis according to the GFR. As the GFR decreases, both the myocardial wall thickness and myocyte width increase. This phenomenon persists even after an adjustment for confounding factors such as age, sex, hypertension, and diabetes mellitus. Similarly, the area of myocardial fibrosis increases from 3.22% to 6.14% as the GFR declines from >60 to <30 mL/min/1.73 m$^2$. The fibrosis itself exhibits patchy and interstitial patterns, and areas of fibrosis are found mainly in the perivascular space. In addition, some severe fibrotic lesions appear to spread into myocardial fiber bundles in the form of pericellular

fibrosis (Figure 2). These findings are fully consistent with reactive fibrosis development. Even after adjustment for confounding factors such as hypertension and diabetes, fibrosis exhibits a tendency to increase as eGFR declines. Additionally, a uremic milieu may also be considered an important pathological condition that may contribute to reactive fibrosis development. Renal anemia, iron deficiency [28], and hyperphosphatemia [29] all may play some role in increasing the area of myocardial fibrosis with decreasing levels of eGFR.

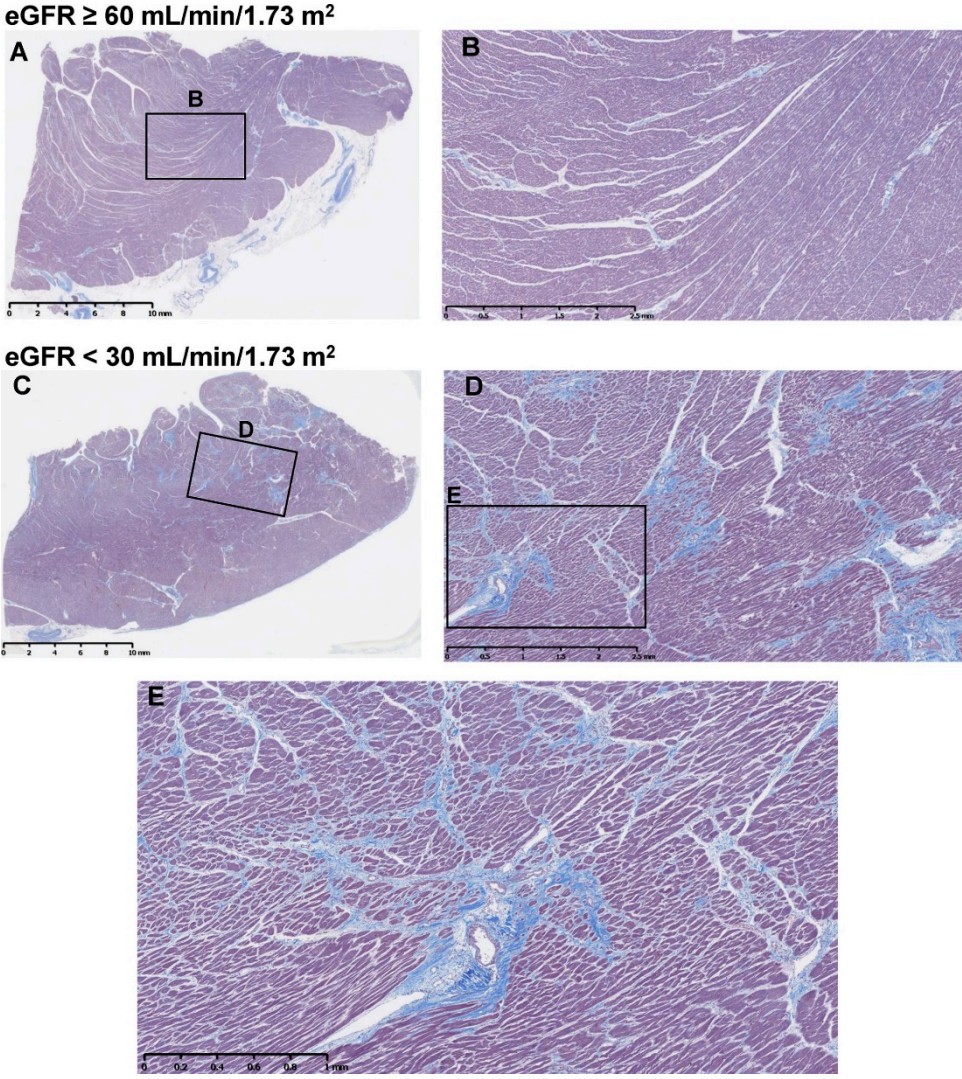

**Figure 2.** Myocardial tissues in samples with estimated glomerular filtration rates (eGFRs) $\geq$60 and <30 mL/min/1.73 m$^2$ (Masson trichrome staining). Reproduced with permission from Izumaru et al., [27], approved by Elsevier. (**A**,**C**) Light microscopic views of myocardial tissues in samples with eGFRs (**A**) $\geq$60 and (**C**) <30 mL/min/1.73 m$^2$. (**B**,**D**) Corresponding lesions from the boxed areas indicated in (**A**,**C**), respectively. (**E**) Corresponding lesion from the boxed area indicated in (**D**), which shows the pattern of patchy and interstitial fibrosis. The area of fibrosis was found mainly in the perivascular space.

## 3. Mechanism of Reactive Myocardial Interstitial Fibrosis

Fibrosis is an essential process in the repair of damaged tissue and wounds. In general, fibrosis occurs when there is an imbalance between collagen synthesis and degradation irrespective of tissue repair. Some pathological factors promote synthesis more than degradation of collagen, resulting in an accumulation of collagenous tissue and leading to organ dysfunction. In pathologically, fibrosis is the excessive deposition of extracellular matrix (ECM), such as collagens and fibronectin [30]. For this process, myofibroblast-

mediated fibrosis in the myocardium is the hallmark of pathophysiological cardiac fibrosis and remodeling [31]. A common feature of cardiac fibrotic diseases is the activation of fibroblasts and their differentiation into myofibroblasts, which express and secrete much higher levels of ECM proteins. Myofibroblasts are not usually present in healthy cardiac tissue, with the exception of heart valve leaflets. However, the transition from fibroblasts to myofibroblasts can be induced by changes in the physical environmental condition of the body. Among them, three factors, mechanical stress, inflammation, cytokines, and growth factors are particularly closely related to the pathogenesis of chronic kidney disease [31].

Pressure loading of the heart, as in aortic valve stenosis and systemic hypertension, is known to increase wall stress and promote reactive fibrosis in the left ventricle chamber. A gradient of fibrosis extends from the endocardial to the epicardial surface. This reflects the gradient in chamber wall stress that results from the increased pressure load. Fibroblasts can also become activated by mechanical stress. This phenomenon can be explained by the fact that fibroblasts sense mechanical forces through mechanosensitive receptors, such as integrins, ion channels, G-protein coupled receptors, and growth factor receptors [15], and activate downstream pathways that promote matrix transcription. As we are well aware, the frequency of hypertension in patients with chronic kidney disease increases with advanced stages and is more frequent in patients with end-stage kidney disease. Another mechanical force for the left ventricle is volume overload, and it is often seen in ESKD patients. In patients with severe aortic regurgitation, a model of left ventricular volume overload, fibrotic remodeling has been extensively documented [32]. However, in contrast to pressure loading, the network of molecular cascades activated by volume overload-induced mechanical stress for cardiac fibrosis has not been characterized [31].

It is well known that the activity of the renin–angiotensin–aldosterone system and sympathetic nervous system activity are increased in patients with chronic kidney disease. These two neurohormonal factors are representative factors that activate fibroblasts and induce differentiation into myofibroblasts [33,34]. Furthermore, diseases or conditions that trigger inflammatory responses, either systemically or locally, can also cause reactive fibrosis to develop. These include obesity, diabetes, metabolic syndrome, infections of the heart, drugs, and radiation. Depending on the stimulus, reactive fibrosis can develop in a relatively homogeneous pattern throughout the myocardium (interstitial fibrosis), or it may be more prominent in the tissue surrounding intracardiac blood vessels (perivascular fibrosis) [35].

## 4. Detection of Myocardial Fibrosis in a Clinical Setting

The best means for properly evaluating myocardial fibrosis is a histological approach. Myocardial biopsy is a highly invasive method that requires hospitalization. Recently, Holmstrom et al. carried out a unique histological study that evaluated the severity of non-ischemic myocardial fibrosis by using heart samples of 1100 autopsies [36]. To achieve noninvasive detection of moderate or severe fibrosis, they sought an association between the severity of non-ischemic cardiac fibrosis and the findings from resting 12-lead electrocardiograms. They considered that QRS widths greater than 100 ms were associated with advanced myocardial fibrosis. Also, negative T waves on aVR becoming shallower and closer to flat were potential candidates for more volume of myocardial fibrosis. They found that T wave depressions of $-0.1$ mV or less in aVR were associated with advanced myocardial fibrosis. There is no further evidence of a link between aVR-T wave changes and myocardial fibrosis, which may limit its use in clinical practice. However, it is of interest that two Japanese cohort studies previously confirmed that the flat or elevation of a T wave on aVR is closely associated with prognosis in dialysis patients [37,38]. In particular, Sato et al. classified patients into four groups according to the T wave of aVR, as shown in Figure 3, and verified the association with prognosis. It can be confirmed that the prognosis is worse in the group with shallower negative T waves [38].

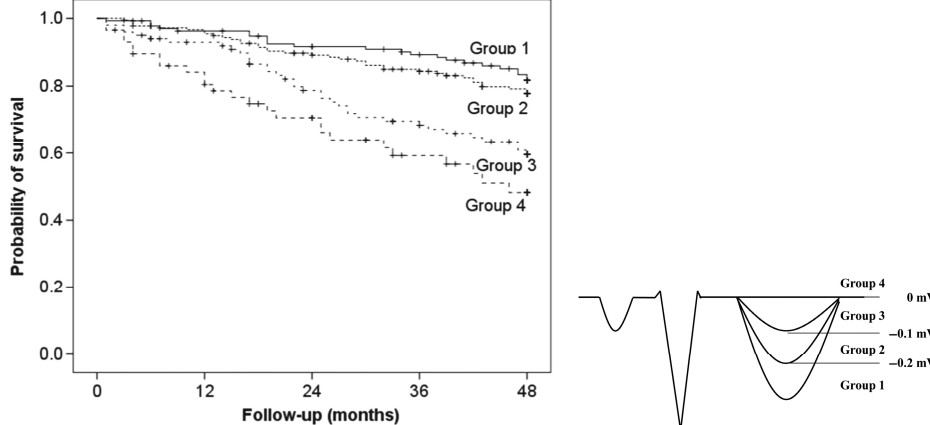

**Figure 3.** Schematic presentation of T-wave amplitude of lead aVR (aVRT)-based groupings. Patients were divided into four groups according to aVRT amplitude as follows: Group 1, aVRT < −0.2 mV; Group 2, −0.2 ≤ aVRT < −0.1 mV; Group 3, −0.1 ≤ aVRT < 0 mV; and Group 4, aVRT ≥ 0 mV. Composite cardiovascular disease (CVD) events-free survival curves according to T-wave amplitude of lead aVR (aVRT) based grouping. ($p < 0.001$, by log-rank test). Ref. [38], approved by Wiley.

Two representative cardiac biomarkers, BNP and troponin T, already used in routine practice have been suggested to be indicators of myocardial fibrosis in CKD patients. Marker of increased LV wall stress, NT-pro BNP, and of myocardial injury, hs-cTnT, were independently associated with native T1, which is evaluated by cardiovascular magnetic resonance and marker for myocardial fibrosis [39]. With regard to possible biomarkers for myocardial fibrosis, two collagen precursors, carboxy-terminal propeptide of procollagen type I and amino-terminal pro-peptide of pro-collagen type III (PIIINP), have been found to correlate with the levels of collagen deposition in dilated cardiomyopathy and in ischemic or hypertensive heart disease [40,41]. Nishimura et al. proposed a cut-off value for PIIINP for cardiovascular events in hemodialysis patients [42]. The cardiovascular-event-free survival rates at five years were lower in patients with serum PIIINP ≥1.75 U/mL, compared with those with that <1.75 U/mL (31.9% vs. 88.2%) (Figure 4). Although it is still unclear whether PIIINP levels are histologically associated with the prevalence of myocardial fibrosis in hemodialysis patients, serum PIIINP might yet be a new biomarker for predicting cardiovascular events in patients undergoing hemodialysis.

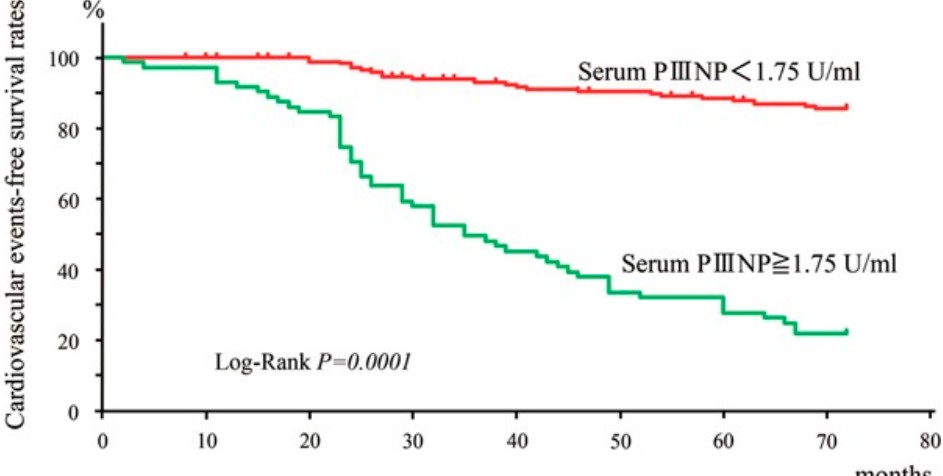

**Figure 4.** Kaplan–Meier analysis of cardiovascular-event-free survival rates for two levels of serum PIIINP concentration (1.75 U/mL) [42]. The cardiovascular-event-free survival rates at 5 years were lower in patients with serum PIIINP ≥1.75 U/mL than in those with <1.75 U/mL (31.9% vs. 88.2%).

The gold standard for the recognition and quantification of cardiac fibrosis is cardiac magnetic resonance using gadolinium-based contrast agents [43]. Gadolinium does not cross the membranes of intact cardiomyocyte cells but is distributed in the extracellular space [44]. A higher distribution volume and slower washout in cases of myocardial scarring or edema results in a hyperintense area in late post-contrast phases when compared to the iso-hypointense normal myocardium [45,46]. By means of such an observation, late gadolinium enhancement can enable the detection of myocardial fibrosis. The association between myocardial fibrosis as measured by LGE and the findings of histological assessments is well-established, showing high sensitivity and specificity [47,48]. However, the use of gadolinium is contraindicated in renal dysfunction patients with GFR levels below 15, including patients on maintenance dialysis, due to concerns about nephrogenic systemic fibrosis. Therefore, a method that does not use gadolinium is required. To this end, the T1 mapping method represents a promising candidate [49,50]. T1 measures the longitudinal relaxation time, which is a specific time constant depending on the tissue composition. Myocardial T1 mapping is a purely parametric technique that physically measures the absolute values of T1 rather than relative differences in signal intensity. Measurements may be taken before (native T1) or after gadolinium administration. Native T1 mapping has been reported as useful in detecting myocardial fibrosis in dialysis patients [51,52].

## 5. Potential Means of Preventing Cardiac Fibrosis

One potential method for retarding the progression of cardiac fibrosis involves interfering with the renin–angiotensin–aldosterone axis. Patients with hypertensive heart disease who were treated using the angiotensin-converting enzyme inhibitor lisinopril [53] exhibited a reduction in the extent of fibrotic deposits, as evaluated by endomyocardial biopsy, as well as an improvement in LV diastolic dysfunction and reduction in LV stiffness. Similar treatment with the angiotensin receptor blocker losartan led to almost identical results [54]. These findings imply that blocking the renin–angiotensin–aldosterone system, along with appropriate control of blood pressure, may be important for the prevention of reactive cardiac fibrosis in cases of hypertensive cardiomyopathy. Myocardial biopsy assessments of heart failure patients have also revealed that the mineralocorticoid receptor antagonist spironolactone reduces areas of myocardial fibrosis [55]. The effect of inhibiting the renin–angiotensin–aldosterone system on prognosis in end-stage renal disease patients is still controversial, with some studies reporting clinical benefit [56], while others report no such benefit [57]. However, a meta-analysis does suggest that aldosterone antagonists reduce the risk of death and morbidity due to cardiovascular and cerebrovascular disease in CKD patients requiring dialysis [58].

In a recent large phase 3 clinical study for patients with heart failure reduced ejection fraction (HFrEF), the risk of hospitalization for heart failure or death from cardiovascular causes was lower among those who received the sodium-glucose cotransporter 2 (SGLT2) inhibitor than among those who received the placebo, regardless of the presence or absence of diabetes [59,60]. It is very interesting that the sub-analysis confirms a significant reduction in the primary endpoint in the SGLT2 inhibitor group, regardless of whether the cardiomyopathy is ischemic or non-ischemic. Furthermore, similar effects have been observed in large clinical trials in heart failure preserved ejection fraction (HFpEF) [61,62]. This suggests that SGLT2 inhibitors may have an inhibitory effect on myocardial fibrosis. It has been shown that the SGLT2 inhibitor empagliflozin reduces myocardial interstitial fibrosis and is linked to improved diastolic dysfunction in diabetic mice [63].

Another possible means to reduce myocardial fibrosis is the loop diuretic torasemide. A study of patients with chronic heart failure suggest that torasemide might inhibit the progression of myocardial fibrosis as an additional benefit to patients receiving standard therapy with RAS-system inhibitors for heart failure [64]. This positive finding has not been reported for furosemide, the most popular loop diuretic in clinical settings. This may be a direct effect of torasemide itself rather than any effect of reduced volume overload in the left ventricle. Lopez et al. found that the activation of the enzyme procollagen type

I carboxy-terminal proteinase (PCP), which catalyzes the extracellular conversion of the precursor procollagen type I into the fibril-forming molecule collagen type I, abnormally increased in the myocardium of heart failure patients [65]. Such activation decreased in torasemide-treated patients and remained unchanged in furosemide-treated patients [65]. Studies of the myocardium in rats found that torasemide, but not furosemide, reduced the expression of aldosterone synthase and transforming growth factor-beta1, both of which can activate PCP in fibrogenic cells [66,67]. A recently published study compared the effects of torasemide and furosemide in patients with diabetes-mellitus-associated heart failure with preserved ejection fraction using changes in the serum levels of C-terminal propeptide of procollagen type I as a biomarker of myocardial fibrosis. No differences in biomarker changes were found between the two treatment groups [68]. The potential use of diuretics in maintenance dialysis patients may be limited; however, torasemide might be a potential treatment option for ESKD patients not on dialysis with volume overload [69]. Further study is needed to confirm the effect of torasemide on cardiac fibrosis in ESKD patients. Experimental studies indicate that several additional antifibrotic therapeutic interventions, including inhibition of fibrosis-promoting inflammatory cytokines such as IL-1 [70], anti-TGF-β approaches [71], and ECM cross-linking enzyme inhibition strategies [72]. Unfortunately, it is undeniable that it will take a long time for these treatments to become available for use in clinical practice.

### 6. Future Directions

To improve the cardiac prognosis of ESKD patients, it is essential to act to prevent myocardial fibrosis or clinically termed HFpEF. However, there are two reasons why their management is very difficult in a clinical setting. One is that there are no laboratory tests, such as blood tests or imaging tests, that can accurately determine the degree of myocardial fibrosis precisely. This means that even if preventive strategies are taken, there is no way to determine their efficacy.

Second, there is still no effective therapy that can retard already advanced myocardial fibrosis. It has been reported that patients with ESKD already have suffered from advanced myocardial fibrosis. This means that it is not possible to regress myocardial fibrosis in patients with ESKD, nor is it expected to improve their prognosis. What can be done today is to first identify the clinical risk factors for myocardial fibrosis. Then, it is suggested that the only preventive approach is to reduce the progression of myocardial fibrosis by strictly correcting these factors. It is important to take these preventive approaches early in the course of CKD.

### 7. Summary

During the course of CKD development, accelerated progression of myocardial fibrosis is caused by well-known factors such as hypertension, diabetes, dyslipidemia, and fluid overload, as well as lesser-known factors associated with the uremic milieu. Diagnostic and therapeutic strategies for reactive myocardial fibrosis are still not established. Because gadolinium is not appropriate for patients with end-stage renal disease, novel diagnostic methods, including biomarkers, are needed. Renin–angiotensin-system inhibitors, SGLT2 inhibitors, and torasemide are candidates for the treatment of myocardial fibrosis, however there is not enough evidence to show that each treatment is effective for myocardial fibrosis in patients with ESKD. To ensure a comprehensive approach to the prevention and management of nonatherosclerotic cardiovascular disease in ESKD patients, multidisciplinary care and collaboration among nephrologists, cardiologists, and other health care providers is necessary.

**Author Contributions:** Original draft preparation, K.N. and N.J.; writing, N.J. All authors have read and agreed to the published version of the manuscript.

**Funding:** This research received no external funding.

**Institutional Review Board Statement:** Not applicable.

**Informed Consent Statement:** Not applicable.

**Data Availability Statement:** No new data were created or analyzed in this study. Data sharing is not applicable to this article.

**Conflicts of Interest:** The authors declare no conflict of interest.

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
