# Peer review of "Non-Ischemic Myocardial Fibrosis in End-Stage Kidney Disease Patients: A New Perspective"

_kidneydial, doi:10.3390/kidneydial3030027_

Round 1

Reviewer 1 Report

The manuscript presents literature review (which should be mantioned in the title), devoted to the important topic – non-ischemic myocardial fibrosis in the patients with advanced CKD. The authors collected and systematically reviewed 68 relevant papers, however, not that many of them were published in 2020-2022, and many – in 1990-1999.

The authors did not include nor fundamental publications of Gerard London, who is one of the top experts in the field (London GM. Left ventricular alterations and end-stage renal disease. Nephrol Dial Transplant. 2002;17 Suppl 1:29-36. doi: 10.1093/ndt/17.suppl_1.29. PMID: 11812909; London, G.M. (2003), The Clinical Epidemiology Of Cardiovascular Diseases In Chronic Kidney Disease: Cardiovascular Disease in Chronic Renal Failure: Pathophysiologic Aspects. Seminars in Dialysis, 16: 85-94. https://doi.org/10.1046/j.1525-139X.2003.16023.x; Zanoli, Luca1; Lentini, Paolo2; Briet, Marie3; Castellino, Pietro4; House, Andrew A.5; London, Gerard M.6; Malatino, Lorenzo4; McCullough, Peter A.7; Mikhailidis, Dimitri P.8; Boutouyrie, Pierre6,9,10. Arterial Stiffness in the Heart Disease of CKD. Journal of the American Society of Nephrology 30(6):p 918-928, June 2019. | DOI: 10.1681/ASN.2019020117), and Nicola Edwards (Edwards, Nicola & Moody, William & Yuan, Mengshi & Hayer, Manvir & Ferro, Charles & Townend, Jonathan & Steeds, Richard. (2015). Diffuse Interstitial Fibrosis and Myocardial Dysfunction in Early Chronic Kidney Disease. The American Journal of Cardiology. 375. 10.1016/j.amjcard.2015.02.015.), nor Di Lullo (Di Lullo L, Gorini A, Russo D, Santoboni A, Ronco C: Left Ventricular Hypertrophy in Chronic Kidney Disease Patients: From Pathophysiology to Treatment. Cardiorenal Med 2015;5:254-266. doi: 10.1159/000435838), and many other important papers.

In the introduction the authors indicate “About 20 years have passed since chronic kidney disease (CKD) was first recognized as a contributor to atherosclerotic cardiovascular disease, resulting in what would be termed cardiorenal syndrome (CRS)”.  As the authors point below, CRS is divided into 5 subcategories, therefore, to avoid confusion, it should be mentioned from the very beginning, not afterwards, that we are talking about CRS type 4. “About 20 years have passed since chronic kidney disease (CKD) was first recognized as a contributor to atherosclerotic cardiovascular disease, resulting in what would be termed cardiorenal syndrome (CRS) type 4”.

References to some publications are presented as: the authors of [13], the authors of [14], the authors of [15]…. That is not polite; the wording should be changed like “Go AS et al…”

Figure 1 referred to number [30], and Figure 3 referred to [35]. Did the authors obtained permission to reproduce these Figures&. If not, Figure 1 and Figure 23 should be excluded from the manuscript.

Beyond that, there are some typos: eg. Prognoses instead of prognosis; Have improved instead of had improved

Thereare some typos, indicated above

Author Response

Dear chief editor, and reviewers,

First of all, we would like to sincerely thank the chief editor and three reviewers for their constructive comments that helped us refine our review article. All of the comments are serious points and we would like to inform you that we have received them seriously and modified them one by one according your comments. Most importantly, the process of deactivating the software in the references resulted in the misplacement of the adopted articles. Thus, we are pleased to inform you that we have carefully reviewed the cited references precisely again. Also, in light of the suggestion that the entire document is redundant, we tried to be more focused and shorter one. We have prepared two files available for you. One keeps a history of changes and is marked in red. The other reflects all of the changes. Thank you again for your kind review.

Reviewer 1

We would like to sincerely thank the reviewer for their constructive comments that helped us refine our review article. All of the comments are serious points and we would like to inform you that we have received them seriously and modified them one by one according your comments. Most importantly, the process of deactivating the software in the references resulted in the misplacement of the adopted articles. Thus, we are pleased to inform you that we have carefully reviewed the cited references precisely again. Also, in light of the suggestion that the entire document is redundant, we tried to be more focused and shorter one. We have prepared two files available for you. One keeps a history of changes and is marked in red. The other reflects all of the changes. Thank you again for your kind review.

The manuscript presents literature review (which should be mentioned in the title), devoted to the important topic – non-ischemic myocardial fibrosis in the patients with advanced CKD. The authors collected and systematically reviewed 68 relevant papers, however, not that many of them were published in 2020-2022, and many – in 1990-1999. The authors did not include nor fundamental publications of Gerard London, who is one of the top experts in the field.

London GM. Left ventricular alterations and end-stage renal disease. Nephrol Dial Transplant. 2002;17 Suppl 1:29-36. doi: 10.1093/ndt/17.suppl_1.29. PMID: 11812909; 

London, G.M. (2003), The Clinical Epidemiology Of Cardiovascular Diseases In Chronic Kidney Disease: Cardiovascular Disease in Chronic Renal Failure: Pathophysiologic Aspects. Seminars in Dialysis, 16: 85-94. https://doi.org/10.1046/j.1525-139X.2003.16023.x;

Zanoli, Luca1; Lentini, Paolo2; Briet, Marie3; Castellino, Pietro4; House, Andrew A.5; London, Gerard M.6; Malatino, Lorenzo4; McCullough, Peter A.7; Mikhailidis, Dimitri P.8; Boutouyrie, Pierre6,9,10. Arterial Stiffness in the Heart Disease of CKD. Journal of the American Society of Nephrology 30(6):p 918-928, June 2019. | DOI: 10.1681/ASN.2019020117), and Nicola Edwards (Edwards, Nicola & Moody, William & Yuan, Mengshi & Hayer, Manvir & Ferro, Charles & Townend, Jonathan & Steeds, Richard. (2015). Diffuse Interstitial Fibrosis and Myocardial Dysfunction in Early Chronic Kidney Disease. The American Journal of Cardiology. 375. 10.1016/j.amjcard.2015.02.015.), nor Di Lullo (Di Lullo L, Gorini A, Russo D, Santoboni A, Ronco C: Left Ventricular Hypertrophy in Chronic Kidney Disease Patients: From Pathophysiology to Treatment. Cardiorenal Med 2015;5:254-266. doi: 10.1159/000435838), and many other important papers.

Thank you very much for your kind comment. We have read it and quoted all of them in the appropriate section.

In the introduction the authors indicate “About 20 years have passed since chronic kidney disease (CKD) was first recognized as a contributor to atherosclerotic cardiovascular disease, resulting in what would be termed cardiorenal syndrome (CRS)”.  As the authors point below, CRS is divided into 5 subcategories, therefore, to avoid confusion, it should be mentioned from the very beginning, not afterwards, that we are talking about CRS type 4. “About 20 years have passed since chronic kidney disease (CKD) was first recognized as a contributor to atherosclerotic cardiovascular disease, resulting in what would be termed cardiorenal syndrome (CRS) type 4”.

Thank you for pointing this out. We have made the changes you suggested. As you indicated, we have inserted an explanation of CRS4 in the opening section.

Page 1, line 20-24 on revised manuscript reflecting corrections.

References to some publications are presented as: the authors of [13], the authors of [14], the authors of [15]…. That is not polite; the wording should be changed like “Go AS et al…”

Thank you for your kind comments. We sincerely apologize for the inappropriate description. We have changed the description to be a politer one.

Figure 1 referred to number [30], and Figure 3 referred to [35]. Did the authors obtained permission to reproduce these Figures? If not, Figure 1 and Figure 2, 3 should be excluded from the manuscript.

Thank you very much for your important comment, we have obtained permission for Figure 2 and 4, and we are aware that Figures 1 and 3 can be cited without permission according to the journal's regulations. Please note that the two articles [30] and [35] you mentioned on original manuscript have been given new numbers, [15] and [39] respectively on revised manuscript reflecting corrections.

Beyond that, there are some typos: eg. Prognoses instead of prognosis; Have improved instead of had improved

We are very sorry about the typos. We have corrected them.

Reviewer 2 Report

Overall, this is very well written manuscript there are several discussions that I would like the authors to additionally highlight. 

One possible improvement for the understanding and treatment of non-atherosclerotic cardiovascular disease in ESKD is to conduct more research into the underlying mechanisms and risk factors of myocardial fibrosis. This can include investigating the role of inflammation, oxidative stress, and endothelial dysfunction in the development of fibrosis, as well as identifying potential biomarkers for early detection and monitoring of fibrotic changes.

Another possible improvement is to develop and implement targeted therapies that can specifically address non-atherosclerotic cardiovascular disease in ESKD patients. For instance, drugs that can inhibit the activation of fibroblasts or reduce the deposition of extracellular matrix components in the myocardium can be explored. In addition, novel treatments that can modulate the immune system or improve microcirculation in the heart may also be beneficial.

Furthermore, it is important to optimize the management of traditional cardiovascular risk factors in ESKD patients, such as hypertension, diabetes, and dyslipidemia, as these conditions can also contribute to the development and progression of non-atherosclerotic cardiovascular disease. This can involve a combination of lifestyle modifications and pharmacological interventions, tailored to the specific needs and comorbidities of each patient.

Finally, there is a need for multidisciplinary care and collaboration between nephrologists, cardiologists, and other healthcare providers to ensure a comprehensive approach to the prevention and management of non-atherosclerotic cardiovascular disease in ESKD patients. This can involve regular monitoring and screening for cardiovascular complications, as well as individualized treatment plans that take into account the unique challenges and complexities of this patient population.

English is well written.

Author Response

Dear chief editor, and reviewers,

First of all, we would like to sincerely thank the chief editor and three reviewers for their constructive comments that helped us refine our review article. All of the comments are serious points and we would like to inform you that we have received them seriously and modified them one by one according your comments. Most importantly, the process of deactivating the software in the references resulted in the misplacement of the adopted articles. Thus, we are pleased to inform you that we have carefully reviewed the cited references precisely again. Also, in light of the suggestion that the entire document is redundant, we tried to be more focused and shorter one. We have prepared two files available for you. One keeps a history of changes and is marked in red. The other reflects all of the changes. Thank you again for your kind review.

Reviewer 2

We would like to sincerely thank the reviewer for their constructive comments that helped us refine our review article. All of the comments are serious points and we would like to inform you that we have received them seriously and modified them one by one according your comments. Most importantly, the process of deactivating the software in the references resulted in the misplacement of the adopted articles. Thus, we are pleased to inform you that we have carefully reviewed the cited references precisely again. Also, in light of the suggestion that the entire document is redundant, we tried to be more focused and shorter one. We have prepared two files available for you. One keeps a history of changes and is marked in red. The other reflects all of the changes. Thank you again for your kind review.

Overall, this is very well written manuscript there are several discussions that I would like the authors to additionally highlight. 

Thank you very much for your review.

One possible improvement for the understanding and treatment of non-atherosclerotic cardiovascular disease in ESKD is to conduct more research into the underlying mechanisms and risk factors of myocardial fibrosis. This can include investigating the role of inflammation, oxidative stress, and endothelial dysfunction in the development of fibrosis, as well as identifying potential biomarkers for early detection and monitoring of fibrotic changes.

Thank you for your important remarks. We have revised the mechanism of myocardial fibrosis based on your constructive comments. We have also added that BNP and Troponin T, which are clinically common markers of cardiac disease, are also useful for detecting myocardial fibrosis.

Page 3-4, line 106-148 on revised manuscript reflecting corrections.

Page 5, line 176-180 on revised manuscript reflecting corrections.

Another possible improvement is to develop and implement targeted therapies that can specifically address non-atherosclerotic cardiovascular disease in ESKD patients. For instance, drugs that can inhibit the activation of fibroblasts or reduce the deposition of extracellular matrix components in the myocardium can be explored. In addition, novel treatments that can modulate the immune system or improve microcirculation in the heart may also be beneficial.

Thank you for your important suggestions. In accordance with your suggestion, we have included additional information on possible treatments for the prevention of myocardial fibrosis.

Page 7, line 267-271 on revised manuscript reflecting corrections.

Furthermore, it is important to optimize the management of traditional cardiovascular risk factors in ESKD patients, such as hypertension, diabetes, and dyslipidemia, as these conditions can also contribute to the development and progression of non-atherosclerotic cardiovascular disease. This can involve a combination of lifestyle modifications and pharmacological interventions, tailored to the specific needs and comorbidities of each patient. Finally, there is a need for multidisciplinary care and collaboration between nephrologists, cardiologists, and other healthcare providers to ensure a comprehensive approach to the prevention and management of non-atherosclerotic cardiovascular disease in ESKD patients. This can involve regular monitoring and screening for cardiovascular complications, as well as individualized treatment plans that take into account the unique challenges and complexities of this patient population.

Thank you for your important remarks. I agree with your points. We have made changes in the conclusion section to add the importance of intervention for classic risk factors and the importance of collaboration between cardiologists and nephrologists to prevent non-ischemic heart disease in patients with end-stage kidney disease.

Page 7, line 278-284 on revised manuscript reflecting corrections.

Reviewer 3 Report

I read the review Non-Ischemic Myocardial Fibrosis in End-Stage Kidney Disease Patients by Kenji Nakata and Nobuhiko Joki.

With their work, the authors aim at highlighting the relevance of the named association, and the possible under-appreciation. Apparently, non-ischemic myocardial fibrosis (NIMF) is most often "reactive" in origin, also due to most common diseases like diabetes, arterial hypertension, and sharing a cause-and-effect relationship with end-stage kidney disease (ESKD). 

While generally interesting, the work at its current stage seems too redundant, and somewhat laborious and vague.  I suggest a much more focused approach. This goes for the whole manuscript. But foremost, all the citations need to be re-checked (see specifics as examples below, but obviously, I have not checked all citations). 

Specifically there are major and minor issues that need resolving, in my opinion.  

Major:

- Please give a quick overview on all sub-categories of the cardio renal syndrome. Citation [3] seems inappropriate ([Article in Croatian]). What about https://doi.org/10.1161/CIR.0000000000000664?

- It comes as a surprise that the authors mention their intention to focus on non-ischemic myocardial fibrosis in end-stage kidney disease (ESKD) in more than one occasion, and yet spend quite some words on just the opposite: ischemic cardiomyopathy and chronic kidney disease (CKD). I am  a strong believer in reliability of the title of a manuscript. Hence, I suggest to have a minimal information acknowledging that ischemic cardiomyopathy plays a significant role in CKD and ESKD, while little is known about non-ischemic myocardial fibrosis, and then to dive deeper into the matter. Specifically, this would mean to adapt the introduction and consequently to remove "2. CKD and Atherosclerotic Cardiovascular Disease " and "3. Non-Atherosclerotic Cardiovascular Disease in ESKD".

- If must stay, please review the paragraph on myocardial infarction. This is somewhat out of context and misleading, e.g. "Type 2 myocardial infarction patients often present with NSTEMI [20]." is neither supported by typical clinical settings (rather the other way around), nor by the citation. But there is more in this paragraph in need of revision. Please refer to the proper guideline definitions (and the proper wording / content of the work cited). 

- This study really is focused on patient with heart failure with preserved ejection fraction (HFpEF) and CKD. The citation "and non-ischemic dilated cardiomyopathy [34]." [DCM], when trying to relate to reactive fibrosis is difficult: (1) this is a study of post-mortem patients, having died from anything but renal disease (2) the validity of the study might be questioned: (a) the number is extremely limited (n=22), (b) there is no information on statistical methods used,  while presenting the data quite unusually with "Mean [SEM]", thus offering a pseudo-precision as opposed to "Mean [SD]", or - even more appropriate for the low number - "Media [IQR]", (3) dilatation was defined by measuring the mitral annular circumference only, but no information is given on the clinical aspects that define DCM (LVEF, LVEDD, LVEDV, etc.)

- Figure 1. is not found in [30], and the citation is not supporting the claim of reactive/diffuse myocardial fibrosis (there is plenty mentioning of replacement fibrosis, though).

- Figure 2 is showing the same scale for panels B, and D. However, looking at the actual image, D seems to have a greater magnification than B. Please re-check.

- Figure 3. is not found in [35]

- Citation [35] seems not complete: "35. (JRDR) TJSfDTRDR: Annual..." 

- Citation [48] is not concerned with ECG (and not "recent" at all, when published in 1995), citation [46], Sato et. al. included patients with ischemia without discrimination, while citation [45] did, but then did not find a significance of the OR for the T-Wave abnormality (0.868 (0.188–4.021), p=0.857). If the diagnostic tool of ECG and aVRT is to remain, stronger evidence and reliable sources are needed. Then, please also add a figure showing what is meant. Also rephrase "elevation" because this is clearly misleading in the context of negative values. 

- Citation [60] seems not complete: "60. <Circulation2000 fibrosis risinopril.pdf>." Is this "Circulation. 2000;102:1388-1393."? If so, please note that at least the aspect of diastolic function improvement should be interpreted with caution (low number of patients). Same goes for [61]. So far only SGLT2-inhibitors (may) play a role in HFpEF patients (compare JAMA. 2023;329(10):827-838. doi:10.1001/jama.2023.2020, with the EMPEROR-preserved study also in need of careful interpretation. The matter remains complicated, I fear).

- "The authors of [69] found...", and also "...were found between the two treatment groups [69]. " - there is no citation [69], and it is also questionable, if both citations [69] mean the same citation, from the content of the text in the manuscript. 

- In the conclusion, "settings, while experimental models have confirmed the antifibrotic effects of the TGF-β-signaling inhibitors pirfenidone and tranilast [68]. " is newly introduced. Please move this to the appropriate section and discuss there in-depth, since the conclusion should only wrap up (the intention of) the review, but not introduce new data.

- What is intended by the last sentence "However, while the quest for improved medication is important, it must be remembered that the best means of preventing reactive myocardial fibrosis is using appropriate management methods that inhibit the progression of CKD and thus prevent end-stage kidney disease."? In the section concerned with the topic, only medical strategies are discussed. Indeed, the section is void of strategies like obesity control, regular exercise, etc., especially when talking about heart failure with preserved ejection fraction. 

Minor: 

- Please remove (seemingly) tendentious claims, such as "The focus of research into cardiovascular disease in patients with end-stage kidney disease is shifting from ischemic to non-ischemic heart disease ", or "Reactive myocardial fibrosis is now a central area of research." (both taken form the summary). What is (a) the proof to the claim, i.e. citation showing the shift in research, and (b) the point? There are other sentences like this, also concerned with other topics, so please re-check. I suggest to either rephrase to a clear message, or to remove. 

- Ishii did not contribute to [16, 17] as first author. Please use common forms of citation mentioning in the manuscript. 

The quality of the english language seems fine, in general. But I am not a native speaker. Yet, I dare to say that in terms of style (redundancy, inflation), there is potential for improvement.  

Author Response

Dear chief editor, and reviewers,

First of all, we would like to sincerely thank the chief editor and three reviewers for their constructive comments that helped us refine our review article. All of the comments are serious points and we would like to inform you that we have received them seriously and modified them one by one according your comments. Most importantly, the process of deactivating the software in the references resulted in the misplacement of the adopted articles. Thus, we are pleased to inform you that we have carefully reviewed the cited references precisely again. Also, in light of the suggestion that the entire document is redundant, we tried to be more focused and shorter one. We have prepared two files available for you. One keeps a history of changes and is marked in red. The other reflects all of the changes. Thank you again for your kind review.

Reviewer 3

We would like to sincerely thank the reviewer for their constructive comments that helped us refine our review article. All of the comments are serious points and we would like to inform you that we have received them seriously and modified them one by one according your comments. Most importantly, the process of deactivating the software in the references resulted in the misplacement of the adopted articles. Thus, we are pleased to inform you that we have carefully reviewed the cited references precisely again. Also, in light of the suggestion that the entire document is redundant, we tried to be more focused and shorter one. We have prepared two files available for you. One keeps a history of changes and is marked in red. The other reflects all of the changes. Thank you again for your kind review.

I read the review Non-Ischemic Myocardial Fibrosis in End-Stage Kidney Disease Patients by Kenji Nakata and Nobuhiko Joki.

Thank you very much for your very courteous review. We have taken your comments seriously and have revised them. Thank you again for your review. Regarding the citation of references, the location of the cited article was misplaced when the software was desynchronized and the English editing was performed. We are very sorry. We have carefully rechecked each one.

With their work, the authors aim at highlighting the relevance of the named association, and the possible under-appreciation. Apparently, non-ischemic myocardial fibrosis (NIMF) is most often "reactive" in origin, also due to most common diseases like diabetes, arterial hypertension, and sharing a cause-and-effect relationship with end-stage kidney disease (ESKD). While generally interesting, the work at its current stage seems too redundant, and somewhat laborious and vague.  I suggest a much more focused approach. This goes for the whole manuscript. But foremost, all the citations need to be re-checked (see specifics as examples below, but obviously, I have not checked all citations). Specifically there are major and minor issues that need resolving, in my opinion.  

Thank you for your interest in peer review. NIMF is still an unexplored area, and we cannot deny the possibility that my personal views may have been too much involved in the content of my writing. According to your suggestion that the entire document is redundant, we tried to be more focused and shorter one.

Major comments

Please give a quick overview on all sub-categories of the cardio renal syndrome. Citation [3] seems inappropriate ([Article in Croatian]). What about https://doi.org/10.1161/CIR.0000000000000664?

Thank you for your comment. We have replaced the article with the one you suggested Circulation 2019, 139(16): e840-e878, and put reference number [3] on revised manuscript reflecting corrections. According to your comment and suggestions from reviewer 1, the text at the beginning of the article has been changed to make the content to be simpler.

Page 1, line 20-24 on revised manuscript reflecting corrections.

It comes as a surprise that the authors mention their intention to focus on non-ischemic myocardial fibrosis in end-stage kidney disease (ESKD) in more than one occasion, and yet spend quite some words on just the opposite: ischemic cardiomyopathy and chronic kidney disease (CKD). I am a strong believer in reliability of the title of a manuscript. Hence, I suggest to have a minimal information acknowledging that ischemic cardiomyopathy plays a significant role in CKD and ESKD, while little is known about non-ischemic myocardial fibrosis, and then to dive deeper into the matter. Specifically, this would mean to adapt the introduction and consequently to remove "2. CKD and Atherosclerotic Cardiovascular Disease " and "3. Non-Atherosclerotic Cardiovascular Disease in ESKD".

Thank you very much for your constructive comments. We agree with your opinion. First, we have deleted paragraphs 2. “CKD and Atherosclerotic Cardiovascular Disease “and 3. “Non-Atherosclerotic Cardiovascular Disease in ESKD” and rewritten the introduction. Thank you very much for your kind suggestions.

Page 1, line 29-40 on revised manuscript reflecting corrections.

If must stay, please review the paragraph on myocardial infarction. This is somewhat out of context and misleading, e.g. "Type 2 myocardial infarction patients often present with NSTEMI [20]." is neither supported by typical clinical settings (rather the other way around), nor by the citation. But there is more in this paragraph in need of revision. Please refer to the proper guideline definitions (and the proper wording / content of the work cited). 

Thank you very much for your suggestions. As you pointed out, we believe the content is outside of the subject matter. We have decided to delete this paragraph in conjunction with the points raised in the previous paragraph.

This study really is focused on patient with heart failure with preserved ejection fraction (HFpEF) and CKD. The citation "and non-ischemic dilated cardiomyopathy [34]." [DCM], when trying to relate to reactive fibrosis is difficult: (1) this is a study of post-mortem patients, having died from anything but renal disease (2) the validity of the study might be questioned: (a) the number is extremely limited (n=22), (b) there is no information on statistical methods used,  while presenting the data quite unusually with "Mean [SEM]", thus offering a pseudo-precision as opposed to "Mean [SD]", or - even more appropriate for the low number - "Media [IQR]", (3) dilatation was defined by measuring the mitral annular circumference only, but no information is given on the clinical aspects that define DCM (LVEF, LVEDD, LVEDV, etc.)

Thank you for your important suggestion. We agree with your opinion that the article [34] is not suitable for adoption as an evidence for reactive fibrosis. We have removed the paper, reference number [37] for revised manuscript.

Page 4, line 178-179 on revised manuscript NOT reflecting corrections.

Figure 1. is not found in [30], and the citation is not supporting the claim of reactive/diffuse myocardial fibrosis (there is plenty mentioning of replacement fibrosis, though). Figure 3. is not found in [35]

The location of the cited article was misplaced when the software was desynchronized and the English editing was performed. We are very sorry. We have carefully rechecked each one. The reference number [30] in the first draft corresponds to [15] on revised manuscript reflecting corrections. Figure 3 in the initial draft will become Figure 4 on revised manuscript reflecting corrections. It is cited from reference number [39].

Figure 2 is showing the same scale for panels B, and D. However, looking at the actual image, D seems to have a greater magnification than B. Please re-check.

Thank you for the comments. We have rechecked the references cited and there is no mistake in the description. Photographs D and B were the same magnified image.

Citation [35] seems not complete: "35. (JRDR) TJSfDTRDR: Annual..." 

We apologize for the lack of confirmation. We have corrected it, and put reference number as [22] on revised manuscript reflecting corrections.

Citation [48] is not concerned with ECG (and not "recent" at all, when published in 1995), citation [46], Sato et. al. included patients with ischemia without discrimination, while citation [45] did, but then did not find a significance of the OR for the T-Wave abnormality (0.868 (0.188–4.021), p=0.857). If the diagnostic tool of ECG and aVRT is to remain, stronger evidence and reliable sources are needed. Then, please also add a figure showing what is meant. Also rephrase "elevation" because this is clearly misleading in the context of negative values. 

First of all, we would like to correct the reference number. The reference number 48 in the first draft corresponds to [33] on revised manuscript reflecting corrections.

As you pointed out, there is no strong evidence to use aVR-guided T-wave amplitude to understand myocardial fibrosis. We have rewritten the statement as follows.

“Also, negative T wave on aVR becoming shallower and closer to flat was potential candidates for more volume of myocardial fibrosis. They found that T wave elevations of -0.1 mV or more on aVR were associated with advanced myocardial fibrosis. There is no further evidence of a link between aVR-T wave changes and myocardial fibrosis, which may limit its use in clinical practice.”

Page 4-5, line 157-161 on revised manuscript reflecting corrections.

In addition, we have cited a schematic illustration of aVR-T waves from a study [35] to help readers better understand the results.

Citation [60] seems not complete: "60. <Circulation2000 fibrosis risinopril.pdf>." Is this "Circulation. 2000; 102: 1388-1393."? If so, please note that at least the aspect of diastolic function improvement should be interpreted with caution (low number of patients). Same goes for [61]. So far only SGLT2-inhibitors (may) play a role in HFpEF patients (compare JAMA. 2023;329(10):827-838. doi:10.1001/jama.2023.2020, with the EMPEROR-preserved study also in need of careful interpretation. The matter remains complicated, I fear).

Thank you for your suggestion. Regarding references [60] and [61] in original manuscript ([50] and [51] on revised manuscript reflecting corrections), we must say that those are reports of a small number of cases and are weak evidence. We have rewritten the text as follows.

“These findings imply that blocking the renin–angiotensin–aldosterone system, along with appropriate control of blood pressure, may be an important for the prevention of reactive cardiac fibrosis in cases of hypertensive cardiomyopathy.”

Page 6, line 225-227 on revised manuscript reflecting corrections.

In addition, the possibility that SGLT2i may inhibit myocardial fibrosis has been added as follows.

“In a recent large phase 3 clinical study for patients with heart failure reduced ejection fraction (HFrEF), the risk of hospitalization for heart failure or death from cardiovascular causes was lower among those who received the sodium-glucose cotransporter 2 (SGLT2) inhibitor than among those who received placebo, regardless of the presence or absence of diabetes [56, 57]. It is very interesting that the sub-analysis confirms a significant reduction in primary endpoint in the SGLT2 inhibitor group, regardless of whether the cardiomyopathy is ischemic or non-ischemic. Furthermore, similar effects have been observed in large clinical trials in heart failure preserved ejection fraction (HFpEF) [58, 59]. This suggests that SGLT2 inhibitors may have an inhibitory effect on myocardial fibrosis. It has been shown that SGLT2 inhibitor empagliflozin reduces MIF and is linked to improved diastolic dysfunction in diabetic mice [60].”

Page 6-7, line 235-245 on revised manuscript reflecting corrections.

"The authors of [69] found...", and also "...were found between the two treatment groups [69]. " - there is no citation [69], and it is also questionable, if both citations [69] mean the same citation, from the content of the text in the manuscript. 

We apologize again for the error in the reference number.

The reference number [69] in the first draft corresponds to [62] on revised manuscript reflecting corrections.

In the conclusion, "settings, while experimental models have confirmed the antifibrotic effects of the TGF-β-signaling inhibitors pirfenidone and tranilast [68]. " is newly introduced. Please move this to the appropriate section and discuss there in-depth, since the conclusion should only wrap up (the intention of) the review, but not introduce new data.

According to your comments, we have moved the description of the TGF-beta from the Summary part, to the “Potential means of preventing cardiac fibrosis” part.

Page 7, line 267-271 on revised manuscript reflecting corrections.

- What is intended by the last sentence "However, while the quest for improved medication is important, it must be remembered that the best means of preventing reactive myocardial fibrosis is using appropriate management methods that inhibit the progression of CKD and thus prevent end-stage kidney disease."? In the section concerned with the topic, only medical strategies are discussed. Indeed, the section is void of strategies like obesity control, regular exercise, etc., especially when talking about heart failure with preserved ejection fraction. 

Thank you for your constructive remarks. Based on your comments and those of the second reviewer, we have changed our statement that preventive measures for myocardial fibrosis should be done through a multidisciplinary approach, as follows.

“To ensure a comprehensive approach to the prevention and management of nonatherosclerotic cardiovascular disease in ESKD patients, multidisciplinary care and collaboration among nephrologists, cardiologists, and other health care providers is necessary.”

Page 7, line 281-284 on revised manuscript reflecting corrections.

Minor comments: 

Please remove (seemingly) tendentious claims, such as "The focus of research into cardiovascular disease in patients with end-stage kidney disease is shifting from ischemic to non-ischemic heart disease ", or "Reactive myocardial fibrosis is now a central area of research." (both taken form the summary). What is (a) the proof to the claim, i.e. citation showing the shift in research, and (b) the point? There are other sentences like this, also concerned with other topics, so please re-check. I suggest to either rephrase to a clear message, or to remove. 

Thank you for your constructive remarks. We have eliminated the description you pointed out from abstract and summary part.

- Ishii did not contribute to [16, 17] as first author. Please use common forms of citation mentioning in the manuscript. 

Thank you. As you indicated, Ishii was one of the co-authors, but not the first author. This has been corrected. However, those articles are eliminated from the revised manuscript.

Round 2

Reviewer 1 Report

We thank the authors for the extensive revision of the original version of the manuscript, and for including of some recommended relevant papers. However, there are still some issues for minor correction:

The title should show that the manuscript presents the review of literature

Lines 9, 26 and 40 – still prognoses instead of prognosis

Nothing beyond indicated typos

Author Response

Dear chief editor, and reviewers,

We would like to thank the chief editor and two reviewers for their constructive comments that helped us on 2nd revised manuscript. All of the comments are important points and we would like to inform you that we modified them one by one according your comments. In addition, there was a problem with the references in the first submission, so we have carefully reviewed them again and made some corrections. We have prepared two files available for you. One keeps a history of changes and is marked in red for second revised manuscript. The other reflects all of the changes. Thank you again for your kind review.

Reviewer 1

We thank the authors for the extensive revision of the original version of the manuscript, and for including of some recommended relevant papers.

Thank you very much for your kind review.

However, there are still some issues for minor correction:

The title should show that the manuscript presents the review of literature

Thank you for your comment. We have changed the title as follows in accordance with your suggestion.

“Non-ischemic myocardial fibrosis in end-stage kidney disease patients; a narrative review”

Lines 9, 26 and 40 – still prognoses instead of prognosis

We apologize for the insufficient confirmation. The correction has been made.

Reviewer 3 Report

I read with high interest the revision of "Non-ischemic myocardial fibrosis in end-stage kidney disease patients" by Kenji Nakata and Nobuhiko Joki.

The authors have made a high effort to address my concerns. Especially the re-arrangement of the citations, and also the revision of the subsection "Mechanism of reactive myocardial interstitial fibrosis" have lifted the quality of the manuscript.

Hence, I see high potential in this work. Once the remaining minor issues have been resolved, publication seems adequate.

Minor:

- Please review the manuscript for the citations again. There are some citations in the newly written paragraph from the "Mechanism of reactive myocardial interstitial fibrosis" that lack conncetion to the reference section ("This reflects the gradient in chamber wall stress that results from the increased pressure load [Frontiers of cardiovascular medicine 2022].", "This phenomenon can be explained by the fact that fibroblasts sense mechanical forces through mechanosensitive receptors, such as integrins, ion channels, G-protein coupled receptors and growth factor receptors [Adv Wound Care (New Rochelle) 2018;7:47–56.]", "In patients with severe aortic regurgitation, a model of left ventricular volume overload, fibrotic remodeling has been extensively documented Circulation 2018;137:184-196]", "However, in contrast to pressure loading, the network of molecular cascades activated by volume overload-induced mechanical stress for cardiac fibrosis has not been characterized Cardiovascular Research (2021) 117, 1450–1488].")

- While the new figure greatly improves understandabiliity for the concept, the ECG description for the aVR T-wave abnormalities is still hard to understand. May I suggest from line 157f: "Also, shallower  negative T waves, i.e. closer to the flat line in aVR was a potential indicator for more myocardial fibrosis. In detail, they found that T wave depressions of -0.1 mV or less in aVR were associated with advanced myocardial fibrosis. ". Would that be acceptable?

- Figure 3 - the one with the 4 Groups of different T-Wave patterns - is not referenced to in the manuscript

- In lines 184f, "amino-terminal propeptide of procollagen type III "is repeated from line 182 and then abbreviated as "(PIIINP)". I see no point in this repetition and I suggest to abbreviate already in line 182 and use the abbreviation in line 184. 

- In line 188, "Figure 3" is referenced. However, I think this should read "Figure 4"

- Please define MIF (Line 244)

As an optional thought: to me it seems, that the statement on inflammation, found in lines 139ff could be used as a general and simple theorem to introduce the concepts made her: inflammation is induced, thus causing fibrosis. 

I you agree to this concept, it might be worthwhile to move this higher up to e.g. at the beginning of line 68, followed by the differentiation of replacement fibrosis after acute myocardial infarction (most certainly causing inflammation) vs. reactive fibrosis, or in line 107, just after "Fibrosis is an essential process in the repair of damaged tissue and wounds.". This could enhance further the understandability, as it is then followed by a typical description of fibrosis found in inflammation and, later on, also examples, e.g. pressure,  and volume overload. 

Small points, best reviewed by or assisted from the proofing section of the journal. 

Author Response

Dear chief editor, and reviewers,

We would like to thank the chief editor and two reviewers for their constructive comments that helped us on 2nd revised manuscript. All of the comments are important points and we would like to inform you that we modified them one by one according your comments. In addition, there was a problem with the references in the first submission, so we have carefully reviewed them again and made some corrections. We have prepared two files available for you. One keeps a history of changes and is marked in red for second revised manuscript. The other reflects all of the changes. Thank you again for your kind review.

Reviewer 3

Comments and Suggestions for Authors

I read with high interest the revision of "Non-ischemic myocardial fibrosis in end-stage kidney disease patients" by Kenji Nakata and Nobuhiko Joki.

Thank you very much for your kind review.

The authors have made a high effort to address my concerns. Especially the re-arrangement of the citations, and also the revision of the subsection "Mechanism of reactive myocardial interstitial fibrosis" have lifted the quality of the manuscript. Hence, I see high potential in this work. Once the remaining minor issues have been resolved, publication seems adequate.

Thank you for your kind support.

Minor:

- Please review the manuscript for the citations again. There are some citations in the newly written paragraph from the "Mechanism of reactive myocardial interstitial fibrosis" that lack connection to the reference section ("This reflects the gradient in chamber wall stress that results from the increased pressure load [Frontiers of cardiovascular medicine 2022]."

As you indicated, the citation of the reference was incorrect. We apologize for the mistakes. The correct name for this literature [Frontiers of cardiovascular medicine 2022] is “Front Cardiovasc Med. 2022 May 6;9:886553. doi: 10.3389/fcvm.2022.886553”. The insertion site for this literature is the site to which the [Adv Wound Care (New Rochelle) 2018; 7: 47–56.] is attached, not the site where it is currently inserted.

"This phenomenon can be explained by the fact that fibroblasts sense mechanical forces through mechanosensitive receptors, such as integrins, ion channels, G-protein coupled receptors and growth factor receptors [Adv Wound Care (New Rochelle) 2018;7:47–56.]"

As mentioned in the previous section, instead of [Adv Wound Care (New Rochelle) 2018;7:47–56.], we will replace them with “Front Cardiovasc Med. 2022 May 6;9:886553. doi: 10.3389/fcvm.2022.886553”.

"In patients with severe aortic regurgitation, a model of left ventricular volume overload, fibrotic remodeling has been extensively documented [Circulation 2018;137:184-196]"

There is no mistake about the insertion site in this literature of [Circulation 2018;137:184-196]. We will reflect this in the reference number.

"However, in contrast to pressure loading, the network of molecular cascades activated by volume overload-induced mechanical stress for cardiac fibrosis has not been characterized [Cardiovascular Research (2021) 117, 1450–1488].")

There is no mistake about the insertion site in this literature of [Cardiovascular Research (2021) 117, 1450–1488]. We will reflect this in the reference number.

While the new figure greatly improves understandability for the concept, the ECG description for the aVR T-wave abnormalities is still hard to understand. May I suggest from line 157f: "Also, shallower negative T waves, i.e. closer to the flat line in aVR was a potential indicator for more myocardial fibrosis. In detail, they found that T wave depressions of -0.1 mV or less in aVR were associated with advanced myocardial fibrosis. ". Would that be acceptable?

Thank you very much for your excellent and appropriate comments to make the contents of this review more understandable. We agree with your suggestion and would like to replace the current sentence with “T wave depressions of -0.1 mV or less in aVR were associated with advanced myocardial fibrosis” you suggested. Thank you very much.

Line 158-159 on 2nd revised manuscript reflecting corrections.

- Figure 3 - the one with the 4 Groups of different T-Wave patterns - is not referenced to in the manuscript

I am very sorry, I have inserted a sentence explaining Figure 3, as follows.

“In particular, Sato et al. classified patients into four groups according to the T wave of aVR as shown in Figure 3, and verified the association with prognosis. It can be confirmed that the prognosis is worse in the group with shallower negative T waves.”

Line 163-165 on 2nd revised manuscript reflecting corrections.

- In lines 184f, "amino-terminal propeptide of procollagen type III "is repeated from line 182 and then abbreviated as "(PIIINP)". I see no point in this repetition and I suggest to abbreviate already in line 182 and use the abbreviation in line 184. 

Thank you for your precise remarks. Correction has been made.

Line 183 on 2nd revised manuscript reflecting corrections.

- In line 188, "Figure 3" is referenced. However, I think this should read "Figure 4"

This is exactly right. Sorry for the lack of confirmation.

Line 188 on 2nd revised manuscript reflecting corrections.

- Please define MIF (Line 244)

 Thanks for pointing out that MIF was an abbreviation for myocardial interstitial fibrosis. We have written it correctly.

Line 244 on 2nd revised manuscript reflecting corrections.

As an optional thought: to me it seems, that the statement on inflammation, found in lines 139ff could be used as a general and simple theorem to introduce the concepts made her: inflammation is induced, thus causing fibrosis. 

I you agree to this concept, it might be worthwhile to move this higher up to e.g. at the beginning of line 68, followed by the differentiation of replacement fibrosis after acute myocardial infarction (most certainly causing inflammation) vs. reactive fibrosis, or in line 107, just after "Fibrosis is an essential process in the repair of damaged tissue and wounds.". This could enhance further the understandability, as it is then followed by a typical description of fibrosis found in inflammation and, later on, also examples, e.g. pressure, and volume overload.

Thank you very much for your very important remarks. I agree with the action to improve the reader's understanding by including a sentence earlier in the article stating that inflammation itself is an important cause of myocardial fibrosis. Finally, we have chosen to add the word "inflammation" to the last sentence of the first paragraph of the “Mechanism of reactive myocardial interstitial fibrosis section”. Do you agree with our proposal?

Among them, three factors, mechanical stress, inflammation, cytokines, and growth factors are particularly closely related to the pathogenesis of chronic kidney disease.

Line 119 on 2nd revised manuscript reflecting corrections.